# Ferri- and ferro-electric switching in spontaneously chiral polar liquid crystals

Jordan Hobbs [1] ✉, Calum J. Gibb [2] & Richard. J. Mandle [1,2]

The recent discovery of spontaneous chiral symmetry breaking has demonstrated the possibility of discovering the exotic textures of ferromagnetic systems in liquid crystalline fluid ferro-electrics. We show that the polar smectic mesophase exhibited by the first molecule discovered to exhibit a spontaneously chiral ferroelectric nematic phase is also helical has a strongly varied textural morphology depending in its thermal history and phase ordering. Electro-optic studies demonstrate that the two spontaneously chiral phases exhibit field-induced phase transitions. For the nematic variant, this process is threshold-less and has no hysteresis, while for the smectic it has a clear threshold and shows hysteresis meaning this phase exhibits pseudo-ferrielectric switching, the first of its kind for ferroelectric nematic like phases. We show that helix formation can be both 1st and 2nd order, but when it is 1st it is accompanied by pre-transitional helix formation extending from the phase boundary into the preceding ferroelectric nematic phase.

Spontaneous symmetry breaking is a general phenomena where a symmetric system will undergo some transition to an asymmetric state where it is no longer invariant under some symmetry operation and has wide-ranging consequences for the natural world such as homochirality in biology[1], autocatalysis in chemistry[2], the Higgs mechanism in physics[3] and even in the understanding of crowd control[4]. Liquid crystals can have various components of order (orientational, translational, bond orientation etc.) and, as such, symmetry breaking can have profound effects on the properties exhibited by the phase[5]. Ferroelectric liquid crystals are notable examples of symmetry breaking within soft matter research, resulting in a variety of phase structures with distinctly different properties. Chiral smectic C liquid crystals (SmC*) form a tilted helical structure where the molecular ordering of dipole moments associated with the chiral centre break mirror symmetry, resulting in an orthogonal polar phase that can be made ferroelectric upon proper confinement[6,7]. Bent core liquid crystals can spontaneously break inversion symmetry forming phase structures containing layer polarisations of various types[8–10] as well as also spontaneously breaking chiral symmetry where the layer polarisation rotates around a helix[11]. The twist bend nematic ($N_{TB}$)[12] and associated smectics[13,14] are other examples of chiral symmetry breaking, forming phases that have degenerate left and right handed heliconic molecular organisation, resulting in phases that are both chiral and locally polar[9,15,16].

In 2017, the ferroelectric nematic phase ($N_F$) phase was discovered[17–19] in which the molecules spontaneously align syn-parallel to each other thus breaking inversion symmetry and resulting in longitudinally polar phases[20–30]. It has been shown that the $N_F$ phase can form twisted domains[31] which can occur spontaneously even with degenerate anchoring orientations[27] to reduce the electrostatic cost of polarisation[27,32]. Further investigations into novel $N_F$ compounds led to the discovery of additional longitudinally polar phases[33–40] with two in particular that demonstrated broken inversion symmetry and also distinct spontaneously chiral heliconic molecular organisation[36,41].

The so called ferroelectric twist bend nematic ($N_{TBF}$) phase[41,42] (also referred to as $^{HC}N_F$[43]) and the polar helical smectic C ($SmC_P^H$) phase[36] both spontaneously form heliconic ordering of the LC molecules with a temperature-dependent pitch around ~400–1000 nm, resulting in selective reflection of visible light. Notably, below the $N_{TBF}$ phase a further phase transition into a tilted smectic phase of unknown structure was observed[41]. This raises the question: does the helix unwind in some fashion to form a non-helical ferroelectric SmC phase, as has been recently discovered[37–39], or does the helix remain, meaning the phase is actually a $SmC_P^H$ phase?

[1]School of Physics & Astronomy, University of Leeds, Leeds, UK. [2]School of Chemistry, University of Leeds, Leeds, UK. ✉e-mail: j.l.hobbs@leeds.ac.uk

In this paper we demonstrate this to be the latter case, showing that this unknown smectic phase is a $SmC_P^H$ phase (we will refer to the $SmC_P^H$ phase in **2** briefly as $SmC_F$ throughout the first section of this article) showing continual existence of the heliconic structure and that the texture and defects of the $SmC_P^H$ phase vary strongly on the phase preceding it. We demonstrate that while both heliconic phases show field-induced phase transitions into their non-helical and non-tilted counterparts, the $SmC_P^H$ exhibits hysteresis of this behaviour, indicating ferrielectric switching, while the $N_{TBF}$ does not. We also show that further transitions types into the $SmC_P^H$ phase are possible and that these are accompanied by pre-transitional helix formation.

## Results

### Materials

Table 1 shows the chemical structures and phases transitions of the materials presented here. Details of chemical synthesis and analysis (NMR, HRMS) can be found in the ESI. All four compounds detailed here contain a 1,3-dioxane ring and so will show isomerisation from the equatorial trans state to the axial trans state just as for the classic ferroelectric mesogen DIO[44]. While for DIO this isomerisation will occur above ~140 °C, the increased size of the molecules here increases the energy barrier for the isomerisation to occur. Our experiments suggest that quickly heating to ~150 °C does not result in a reduction in transition temperatures, but sustained time above this temperature results in a reduction in all transition temperatures, indicative of isomerisation. Compounds **1**[36] and **2**[41] have been characterised previously and so their characterisation via Differential Scanning Calorimetry (DSC), X-ray, and Ps measurements are included in the ESI, while compounds **3** and **4** are discussed in the final section of this manuscript.

### Polarised optical microscopy

Compound **1** melts into the $SmA_F$ phase and in cells prepared with either parallel or anti-parallel rubbed surfaces, forms blocky textures when viewed using polarised optical microscopy (POM) (Fig. S1a). From azimuthal rotation of the cell with respect to the polarisers, it is clear that most of the blocky domains are orientated with the director close to the rubbing direction but with some slight in plane $\phi$ rotation diving the texture into domains. The blocky texture results from the restriction of splay deformations by the electrostatic cost of polarisation splay and the restriction of bend deformation by the formation of layers[33,45]. Heating into the SmA phase retains these domains (Fig. S1b), but they are lost upon reaching the N phase where a uniform planar texture is obtained. Upon cooling back into the $SmA_F$ phase a monodomain texture is obtained (Fig. S1c) regardless of cell thickness. No evidence of director twist[27,31,46] is found even in cells using anti-parallel rubbed planar alignment layers.

Immediately upon transitioning to the $SmC_P^H$ phase, striations appear parallel to the rubbing direction with an average periodicity of around 1.2 μm at 80 °C (Fig. 1). While the average periodicity of these striations is generally independent of cell thickness, the periodicity takes a greater possible range of values for increasing cell thickness. Upon cooling, these striations behave quite differently depending on cell thicknesses. For thin cells (1.6 μm) the striation periodicity decreases to around 0.8 μm by crystallisation at around 55 °C (Fig. 1). For thicker cells (5 μm) the striations initially decrease in periodicity before larger less uniformly shaped striations grow in to become the dominant texture (Fig. 1b). For 10 μm thick cells the non-uniform striations grow into thick and distinct defect lines, though these do not have to encompass a full domain and do not show alternative optical activity upon slight de-crossing of the polarisers (Fig. 1c). 20 μm thick cells behave similarly to the 10 μm thick cell, except the temperature threshold for the defect lines is much closer to the $SmA_F - SmC_P^H$ transition (Fig. 1d). These striations are clearly distinct from alternating polarisation domains[36] and form even when a monodomain $SmA_F$

texture aligned under an in-plane electric field is cooled into the $SmC_P^H$ phase. They are also retained upon applying a weak in-plane electric field parallel to the polarisation direction.

The smectic phase, $SmC_F$, exhibited by **2** shows quite different behaviour where in cells of 1.6, 5, 10, and 20 μm thickness the texture is consistently grainy and nondescript (Fig. 1e) when the sample is cooled from the preceding $N_{TBF}$ phase resulting in significant light scattering and low light transmission though the cell (Fig. 1i). The $SmC_F$ phase in compound **2** is monotropic and the texture exhibited by the phase on melting directly from the crystal phase is distinctly different (Fig. 1f). The melt texture is paramorphotic of the crystal phase but does not show any of the defects and scattering found in the phase on cooling. On heating the melted $SmC_F$ texture to ~10 °C below the transition into the $N_{TBF}$ the defects found on cooling begin to form and the transmission through the sample drops significantly, but does decrease to the values obtained on cooling. While selective reflection is difficult to be observed on cooling a faint blue colour can be seen in the $SmC_F$ phase with the naked eye on heating, suggesting that this phase does retain the helical structure of the preceding $N_{TBF}$ phase.

If the $SmC_F$ phase is helical than the differences in optical textures on heating/cooling could result from the fact that the greatest change in pitch length (i.e., likely close to the phase transition) occurs in the same temperature regime as the scattering texture forms. These defects would then be the result of plastic deformation of the pitch though contraction and dilation[47]. However, so far the scattering texture have precluded optical measurement of any pitch length in the $SmC_F$ phase. We also note that the scattering behaviour of **2** in the $SmC_F$ phase is not isolated to cell confinement geometries and shows the same highly scattering and opaque appearance when filled into capillaries or as bulk samples in droplet form or in vials, demonstrating that it is not a result of confinement.

### Binary phase diagram between 1 and 2

Construction of binary phase diagrams is a well-established technique in liquid crystal science to demonstrate whether two materials posses the same phase by establishing miscibility between the two materials across the full phase diagram[48] and has been well applied to the $N_F$ phase already[49]. Binary mixtures of **1** and **2** were created and the resulting phase diagram shown in Fig. 1j. From the binary phase diagram of **1** and **2** the $SmC_P^H$ phase is completely miscible with the $SmC_F$ phase demonstrating that the $SmC_F$ phase is heliconic that the two phases are structurally the same. The studies presented so far have shown the $SmC_F$ found in **2**[41] is the same as as the $SmC_P^H$ phase reported by us[36] and so we shall refer to the $SmC_F$ phase of **2** as a $SmC_P^H$ phase from here on.

The apolar-polar transition temperature (whether this is $N-N_S$ or $SmA-SmA_F$) is roughly linear with concentration while the transition to helical order, whether this is nematic of smectic type, is actually stabilised with a peak at around 30% of **1** for both the $N_{TBF}$ and the $SmC_P^H$ phase indicating that the molecular origins of ferroelectric order vs spontaneously chiral differ for these systems. It also points to the possibility of using mixtures in the future to construct systems with wide phase existence for application purposes. We note that the peak of chiral phase stability also roughly coincides with the point at which the transition pattern switches between $N_F - N_{TBF} - SmC_P^H$ to $SmA_F - SmC_P^H$. It is also this point where the texture and scattering behaviour changes from the highly scattering uncharacteristic texture to the striated texture. Seemingly, the key difference is the fact that the $SmC_F$ texture on cooling is from the $N_{TBF}$ phase, while the $SmC_P^H$ texture is cooled from a $SmA_F$ phase.

The formation of the $SmC_P^H$ phase clearly differs depending on the phase preceding it and here we speculate on the origin of the difference in textures. There could be a significantly larger change in pitch at the $N_{TBF}$–$SmC_F$ transition vs the $SmA_F - SmC_P^H$, though this feels unlikely due to our expectation that the temperature dependence of

**Table 1 | Transition temperatures (T) and associated enthalpies of transition (ΔH) for compounds 1–4 determined by DSC on cooling at a rate of 10 °C min⁻¹**

| No. | | Melt | $SmC_P^H$-$SmA_F$ | $SmA_F$-SmA | SmA-N | $SmC_P^H$-$N_F$ | $SmC_P^H$-$N_{TBF}$ | $N_{TBF}$-$N_F$ | $N_F$-$N_S$ | $N_S$-N | N-Iso |
|---|---|---|---|---|---|---|---|---|---|---|---|
| 1 | T (°C) | 100.9 | 90.1ᵃ | 129.7 | 154.3 | – | – | – | – | – | 225.6 |
| | ΔH (kJ/mol) | 31.1 | – | 0.6 | 0.1 | – | – | – | – | – | 1.0 |
| 2 | T (°C) | 64.0 | – | – | – | – | 91.6 | 104.4ᵃ | 142.6 | 146.4 | 225.1 |
| | ΔH (kJ/mol) | 19.0 | – | – | – | – | 0.1 | – | 0.4 | 0.01 | 1.1 |
| 3 | T (°C) | 93.8 | – | – | – | 90.6 | – | – | 178.4 | 179.5ᵃ | 213.7 |
| | ΔH (kJ/mol) | 19.2 | – | – | – | 0.6 | – | – | 0.7 | – | 1.3 |
| 4 | T (°C) | 99.8 | – | – | – | – | – | – | 119.3 | 126.7 | 233.8 |
| | ΔH (kJ/mol) | 25.1 | – | – | – | – | – | – | 0.2 | 0.01 | 1.6 |

ᵃ determined via POM observations as no corresponding peak was observed in DSC.

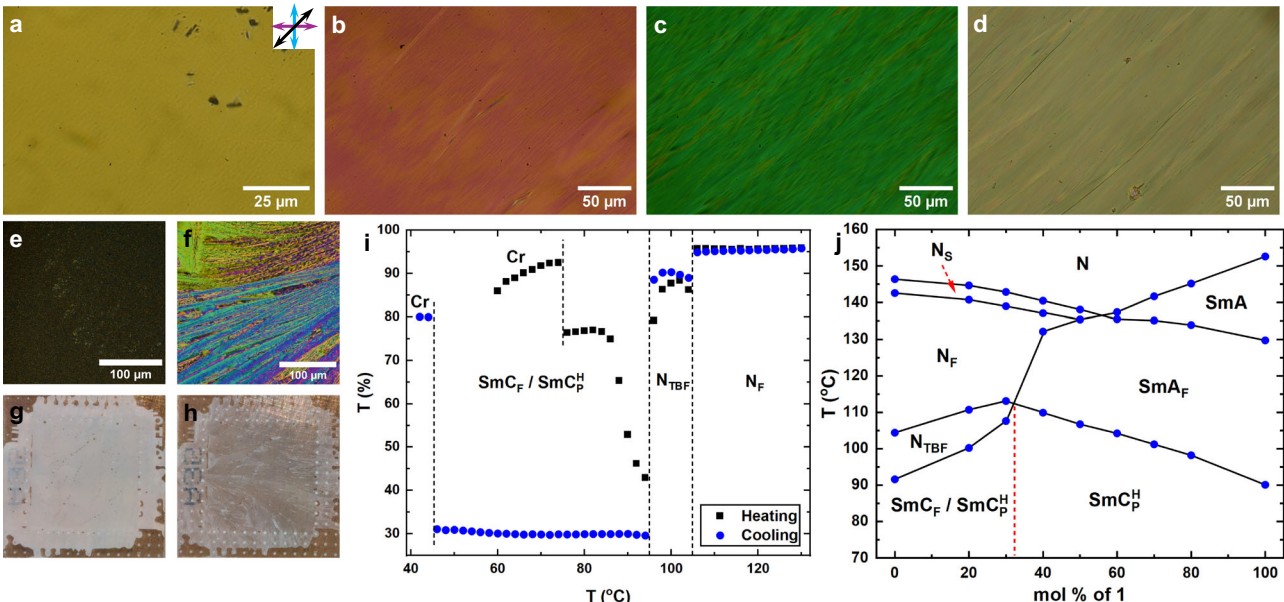

**Fig. 1 | Optical properties of the compound 1 and 2 and phase diagram between them.** POM images taken for compound **1** at 80 °C filled into parallel rubbed, planar aligned cells with a cell gap of **a** 1.6 μm, **b** 5 μm, **c** 10 μm and **d** 20 μm. POM images of **2** at 80 °C filled into parallel rubbed, planar aligned cell with a cell gap of 5 μm on **e** heating directly after melting from the crystal phase and **f** subsequent cooling after heating to the nematic phase. Corresponding macroscopic texture of an entire 4 μm cell filled with **2** at 80 °C on **g** heating and **h** cooling. The brown background colour comes from the hot stage plate. **i** Transmission intensity of white light through a sample of **2** in a 5 μm thick planar aligned parallel rubbed cell. **j** Binary phase diagram between **1** and **2** with 100% compound 1 on the right and 100% compound 2 on the left. The red vertical dashed line marks the point where the $SmC_P^H$ phase switches polymorphism from the highly scattering to the striated, as discussed in the text. For all POM images polarisor axis' and rubbing direction are marked by the blue, purple and black arrows, respectively.

the pitch at the $SmA_F - SmC_P^H$ transition will at least be somewhat similar to that at the $N_F–N_{TBF}$ transition i.e., a steep temperature dependence[41]. We do note a very slight unwinding of the helix just before the $N_{TBF}–SmC_F$ transition (Fig. S13). This is different from the much more significant unwinding of the helix observed for the transition from an $N_{TBF}$ phase to a non-helical $SmC_P$ phase[42] where the helix unwinds critically.

An alternative is that forming layers in a system with an existing helical structure could result in significant defect formation due to potential deformation of the helix to form the regularly spaced smectic layers, although this does not seem to occur for the apolar twist bend nematic and associated smectic phases[14,50,51], though the much smaller pitch, lack of polarity and vastly different origins (electric vs elastic) make this a less than optimal comparison.

Another alternative is that the mechanism of phase structure formation in the $SmC_P^H$ could vary significantly depending on the phase preceding it. This will mostly likely be true to some extent as observation of the temperature-dependent tilt implies that the transition to the $SmC_P^H$ phase from the $SmA_F$ is only 2nd order while the $N_{TBF}–SmC_F$ transition is weakly first-order, however, whether this explains the differences in textures vs some alternative pitch defect effect remains to be determined. It could also be reminiscent to the formation of chevron defects in the $SmC^*$ phase where cooling from a SmA forms chevron structures due to surface pining[52] which can be somewhat reduced when formed from a De Vries SmA phase where the randomly orientated molecular tilt reduces chevron formation[53].

### Electro-optics and ferrielectric switching
**Current response measurements.** The current response of the $SmC_P^H$ of **1** in a 4 μm cell with out of plane bare ITO electrodes is not a single peak, indicating complete reversal of the polarisation. Instead, the current response shows a small peak pre-polarity reversal and a larger peak post-polarity reversal (Fig. 2a). As such, in the original publication[36], the phase was given the subscript designation of "**P**" to indicate polar, as the exact nature of the polarity was unclear from those initial measurements. We suggested that the small peak is associated with the emergence of molecular tilt. From the ground state at $E = 0$ V/μm, increasing the electric field results in the first large peak corresponding to the full reversal of the polarisation vector from a $SmC_P^H$ state to a non-tilted, non-helical $SmA_F$ state. The small peak is then the reformation of the tilt and helix. The peak cannot be due to variation in the helical pitch length as this would not modulate the polarisation beyond a small variation due to changes in the region of partial pitch that may form due to the degenerate anchoring condition of the bare ITO electrode. Variation in the tilt angle, however, would modulate the effective polarisation observed upon switching. Throughout this next section, we shall refer to the smaller peak as the "tilt" peak and the larger one as the "polarisation reversal" peak.

The following discussion is given in context of the current response corresponding to a negative to positive application of electric field. Perhaps unsurprisingly, the $SmC_P^H$ phase of **2** shows a similar current response to that of **1** in terms of a tilt peak and a polarisation reversal peak (Fig. 2b), although the response's do differ in several ways. For **1** the tilt peak initially occurs at a more negative electric field (0.43 V/μm) than for **2** (0.1 V/μm) and is distinctly separate from the polarisation reversal peak. For **2** the peak appears the grow out of the main reversal peak seen for the $N_{TBF}$ phase as it moves to increasingly negative voltages. We interpret this as a consequence of their different phase transitions. Presumably, for **1** in the $SmC_P^H$ phase, the viscosity is similar to other smectic phases even close to the $SmA_F - SmC_P^H$ transition, while for **2** the viscosity close to the $N_{TBF} - SmC_P^H$ transition is lower, having just transitioned from the lower viscosity nematic phase. For both samples, the peaks develop very similarly. Both the tilt peak and the polarisation reversal peak broaden and move to longer time-scales, though this timescale shift is much more apparent for the polarisation reversal peak, which may indicate that the tilt peak is less affected by the viscous-dynamics of the phase.

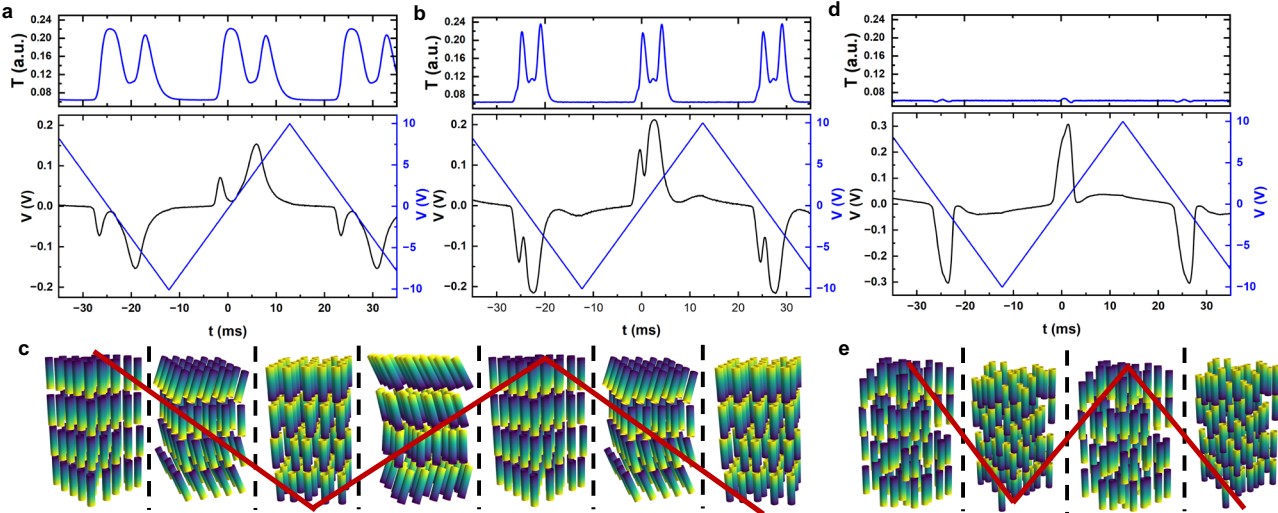

**Fig. 2 | Current response of compounds 1 and 2 in their helical phases.** Electro-optic and current response to an 20 Hz, 10 $V_{RMS}$ applied triangle wave for **a** compound 1 at 75 °C, **b** compound 2 at 75 °C. **c** shows a schematic diagram for the stationary structures formed within the switching process of compound 1 and 2 in the $SmC_P^H$ phase. **d** Electro-optic and current response to an 20 Hz, 10 $V_{RMS}$ applied triangle wave for compound 2 at 99 °C. **e** Shows a schematic diagram for the stationary structures formed within the switching process of compound 2 in the $N_{TBF}$ phase. Each compound is filled into a 4 μm thick cell with bare ITO electrodes. The top portion of the graphs shows the transmission through the cell placed with between crossed polarisers as detected by a photo-diode while the bottom portion is the current response. Within each schematic diagram the red line indicates the applied voltage.

The electro-optic response was measured simultaneously to the current response by measuring the transmission of 589 nm light through the sample set between crossed polarisers (Fig. 2). Again, as for the current response, there are similarities and differences. Both samples show changes in the transmission that couple to the tilt peak or the polarisation reversal peak prohibiting either peak being predominately due to ion flow and accumulation. The electro-optic response is horn like for both samples, although clearly much sharper for **2** indicative of the lower field thresholds for the optical changes, resulting in a narrowed peak. The horn shape itself can be understood by considering that transmission through a birefringent LC slab is given by:

$$\frac{T}{T_o} = \sin^2 2\phi \sin^2\left(\frac{\pi\Delta nd}{\lambda}\right) \qquad (1)$$

where $T/T_o$ is the transmission ratio, $\phi$ is the angle between director and polarisor, $\Delta n$ is the birefringence, $d$ the cell thickness and $\lambda$ the wavelength of light. In the $SmA_F$ state at high electric field, the transmission is a minimum as the sample has homeotropic alignment and the effective birefringence of the sample is close to zero. Reducing the electric field leads to reformation the tilt and regaining of the $SmC_P^H$ state. The out of plane field means the homeotropic alignment is retained, and so the effective birefringence is low, leading to high transmission. As the field moves towards 0 V/μm the homeotropic alignment is lost, resulting in an increased birefringence and a loss in transmission. Increasing the field beyond the polarisation reversal peak results in a flip of the molecules and a field-induced phase transition into the $SmA_F$ state, reducing the birefringence to zero. It does not seem that the polarisation reversal motion from a $SmC_P^H$ state to $SmA_F$ state involves an intermediate $SmC_P^H$ state with reversed polarisation.

For the $N_{TBF}$ phase the current response does not show a tilt peak and only shows a single polarisation reversal peak (Fig. 2c). However, it has been shown that the tilt in the $N_{TBF}$ phase can be manipulated[41] making the lack of an obvious tilt peak surprising. This will be revisited later in the manuscirpt. The electro-optic response is also strange with no significant change in transmission as a function of the applied triangle wave. We expect that at high electric field the molecules are in

the homeotropic state giving low transmission, where only very small fields are required to maintain this state. At low fields the tilt is regained along with the helix, resulting in reduced transmission, which counteracts the increased transmission due to increased birefringence and loss of the homeotropic state.

**In-plane electro-optics of the $SmC_P^H$ phase.** To confirm that the tilt peak is indeed associated with the reformation of molecular tilt POM observation and birefringence measurements of the in-plane switching mechanism is used. The birefringence was measured in the centre of the electrodes using a Berek Compensator with further details given in the SI. Due to the strong formation of defects in the $SmC_P^H$ of **2** we instead will mostly focus on the $SmC_P^H$ of **1. 1** was filled into parallel rubbed 5.5 μm thick cell with two in-plane electrodes with a gap of 100 μm. The rubbing direction was parallel to the direction of the electric field across the electrode gap.

Applying a weak electric field (0.225 V/μm) opposite to the initial director orientation results in the formation of elongated diamond-shaped domains of reversed director (Fig. 3a). These domains grow perpendicular to the applied field until the entire active area is filled (0.3 V/μm). They can extend beyond the active area similar to the $N_F$ phase and in direct contrast to that of the $SmA_F$ phase where polarisation reversal is confined to the active area[33] (Fig. 3a). This phenomena is likely due to the reduced field threshold for polarisation reversal of the $SmC_P^H$ phase vs the $SmA_F$ phase presumably allowing the resulting depolarisation field to propagate beyond the active area. We note that the diamond domains of reversed polarisation that form outside the active area tend to form on the side opposite to the direction of the applied field, though this does not mean that cannot form in the region in the direction of applied field. Increasing the field beyond the switching threshold sees a continuous gain in birefringence and a removal of the striations (Fig. 3a). The final texture is uniform in colour with a birefringence value similar to that of the $SmA_F$ phase at the $SmA_F - SmC_P^H$ transition. Flickering undulations in the texture can be observed, indicating some hydrodynamic flow due to the applied field, but these do not have any obvious pattern to them. The edges of the active area show slightly different birefringence colours due to the non-uniformity of the field in those areas. Removal

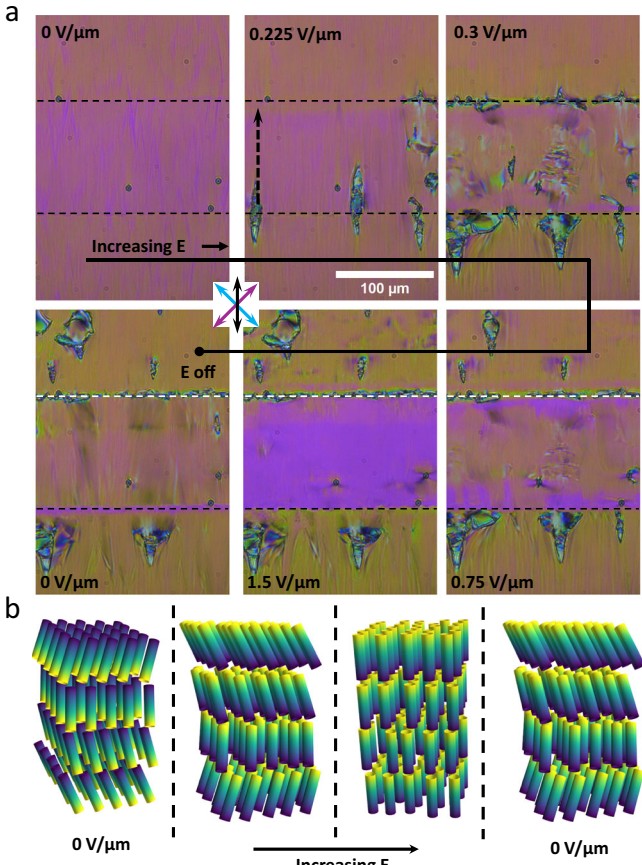

**Fig. 3 | Inplane electro-optics of 1 with an anti-parallel field. a** POM images of compound **1** at 75 °C in the SmC$_P^H$ phase in an IPS cell under applied field. The horizontal dashed lines indicate the limits of the active area, while the vertical dashed arrow indicates the electric field direction. The cell is 5.5 μm thick with parallel rubbed planar alignment. The gap between the two electrodes is 100 μm and the DC field is initially applied anti-parallel the the initial polarisation state. For all POM images, polarisor axes and rubbing direction are marked by the blue, purple and black arrows, respectively. **b** Schematic diagrams of the different states in the switching profile are shown for the IPS cell as described in the text. Only half the pitch length is shown for the helical states.

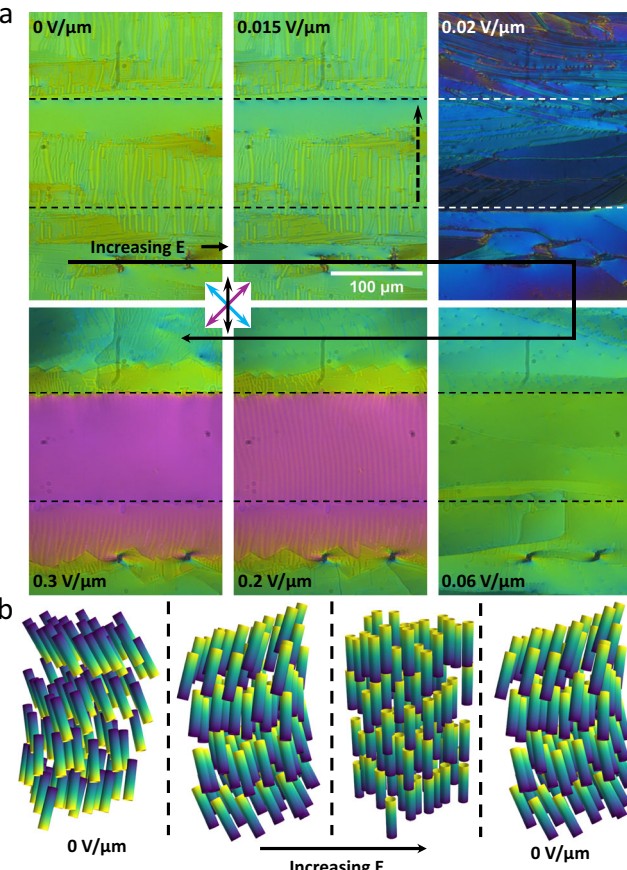

**Fig. 4 | Inplane electro-optics of 2 with an anti-parallel field. a** POM images of compound **2** at 100 °C in the N$_{TBF}$ phase in an IPS cell under application of a DC electric field. The horizontal dashed lines indicate the limits of the active area, while the vertical dashed arrow indicates the electric field direction. The cell is 5.5 μm thick with parallel rubbed planar alignment. The gap between the two electrodes is 100 μm and the DC field is initially applied anti-parallel the the initial polarisation state. For all POM images polarisor axes and rubbing direction are marked by the blue, purple and black arrows, respectively. **b** Schematic diagrams of the different states in the switching profile are shown for the IPS cell as described in the text. Only half the pitch length is shown for the helical states.

of the field results in a texture that is reminiscent of the initial texture though some of the reversed polarisation domains remain outside of the active area as well as a domain line along the edges of the active area, indicating the different polarisation domains. The complete process is shown in Fig. 3b. The sample starts in the SmC$_P^H$ state. Further increase of the electric field reorients the polarisation direction while leaving the helix intact (though we do not speculate on any changes to the helical pitch here). Further increasing the field causes a transition into the non-tilted SmA$_F$ state. Removal of the field reforms the tilt and helix.

**In-plane electro-optics of the N$_{TBF}$ phase.** The current response of the SmC$_P^H$ phase clearly shows a second peak, which we have shown to be related to tilt reformation, indicating that the larger polarisation reversal peak also contains a field-induced phase transition back into the SmA$_F$ phase. The N$_{TBF}$ phase is also heliconical, but its current response does not show a second peak to associate with any field-induced transition. To investigate whether full tilt removal is still possible, we probe the N$_{TBF}$ phase via in-plane switching observations in the same 5.5 μm thick parallel rubbed cells as for the SmC$_P^H$ phase.

The transition to the N$_F$ phase to the N$_{TBF}$ phase is clear by the formation of stripes parallel to the rubbing direction (Fig. 4a). As the

temperature is decreased, these stripes split along the rubbing direction into regions of stripe domains and uniform texture. Since these stripes are parallel to the rubbing direction and thus parallel to the helicoidal axis, they are not pitch repeat units. These are also not regions of opposite polarity as they remain under application of a moderate in-plane dc field and a strong field removes them without evidence of polarisation reversal.

For the N$_{TBF}$ phase the direction of the in-plane polarity can be switched by a dc in-plane electric field, just as for the N$_F$[21,31,54,55] although the switching is somewhat different. At low dc fields (below ~0.02 V/μm) there is a small reduction in birefringence without significant disruption of the texture (Fig. 4a), consistent with the N$_F$ phase where this behaviour has been associated with increased director twist[55]. At a critical voltage (~0.02 V/μm), there is a significant reduction in birefringence and the sample breaks into many domains (Fig. 4a). Slight uncrossing of the polarisers does not reveal any difference in optical activity of these domains. As the voltage is increased further (up to 0.06 V/μm) these domains coalesce and there is a steady increase in birefringence up to the amount of the unperturbed N$_{TBF}$ state with no signs of the undulations found in that unperturbed state (Fig. 4a). Further increase in the voltage sees an increase in the birefringence beyond the unperturbed state and eventually (above 0.15 V/

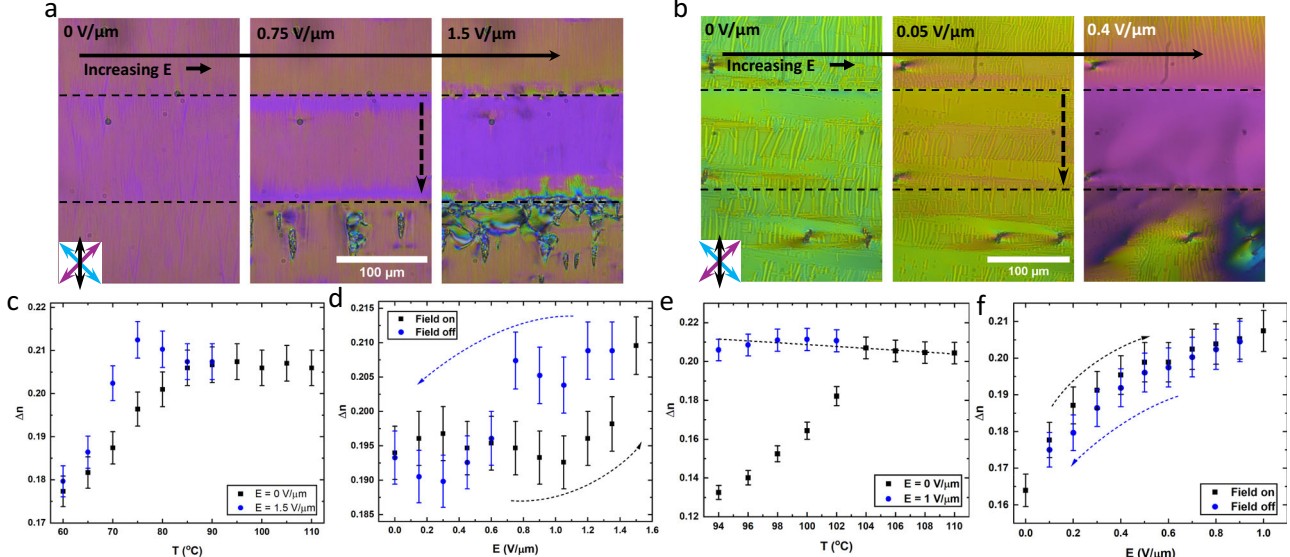

**Fig. 5 | Inplane electro-optics of 1 and 2 with a syn-parallel field.** POM images of **a** compound **1** at 75 °C in the SmC$_P^H$ phase and **b** compound **2** at 100 °C in the N$_{TBF}$ phase both an IPS cell under application of a DC electric field where the cell is 5.5 µm thick with parallel rubbed planar alignment. Here, the DC field is applied syn-parallel to the polarisation direction. For all POM images polarisor axis' and rubbing direction are marked by the blue, purple and black arrows, respectively. Temperature dependence of the birefringence of **c 1** and **e 2** both with and without an applied DC field. Electric field dependence of the birefringence of **d** compound **1** at 75 °C and **f** compound **2** at 100 °C, indicating the hysteresis of the tilt removal for the SmC$_P^H$ phase and the lack of hysteresis for the N$_{TBF}$ phase.

µm) the undulations return before finally (around 0.3 V/µm) obtaining a uniform N$_F$ like texture with a birefringence similar to that of the N$_F$ phase just before transition to the N$_{TBF}$ phase. Just as for the SmC$_P^H$ phase, the effect of the in-plane field can extend beyond the active area. However, due to the lower threshold for switching in the N$_{TBF}$ phase, the full polarisation reversal spreads significantly beyond the active area and it is just the increase of birefringence that is limited to the out of active area region opposite the applied field. The complete process of the N$_{TBF}$ phase is shown in Fig. 3b. The sample starts in the N$_{TBF}$ state where increasing the electric field reorients the polarisation direction while leaving the helix intact. Further increasing the field causes a transition into the non-tilted N$_F$ state. Removal of the field reforms the tilt and helix. It is clear that both the SmC$_P^H$ and N$_{TBF}$ phases can undergo a field-induced phase transition the non-tilted state, which by necessity would remove the helix entirely.

**Ferrielectric switching in the SmC$_P^H$ phase.** Applying a electric field anti-parallel to the polarisation axis of both the SmC$_P^H$ and N$_{TBF}$ phases results in full reversal of the polarisation direction. Further increase in the field results in an increase in birefringence, where this second stage is due to the reduction in molecular tilt. If the electric field is applied syn-parallel to the polarisation direction, no reversal in the polarisation direction is observed. However, an increase in birefringence, associated with the tilt removal, is still observed for both phases (Fig. 4). If the field is removed the tilt is reformed and a drop in birefringence back to the original value is observed. For the SmC$_P^H$ phase of **1**, full tilt removal could only be completed in a specific temperature window close to the SmA$_F$ - SmC$_P^H$ transition (Fig. 5a) due to an increase in field threshold for this effect. Presumably, if higher electric fields were achieved, either through higher voltages or a narrower active area, full tilt removal could be completed for the entire phase window. For the N$_{TBF}$ phase of **2** full tilt removal could be completed for the entire phase (Fig. 5b). We also note that if an in-plane field is applied to the SmC$_P^H$ phase of **2** a uniform birefringent texture could be obtained (Fig. S3) which reverted back to the defect texture upon removal of the field indicating that tilt removal is possible here too. The hysteresis of tilt removal and switching behaviour is clear also in out of plane

measurements (Fig. S4). where tilt is clearly recovered only at lower field strength values than it is lost.

To investigate the threshold and field dependence of the tilt removal, we measure birefringence as a function of applied field. This is measured in the very centre of the active area where the in-plane field is most parallel to the cell plane. For the SmC$_P^H$ phase of **1**, we see that the birefringence only increases after some critical value, where it sharply increases up to a value slightly beyond that of the untilted SmA$_F$ phase. We attribute this increase beyond the SmA$_F$ value either due to a increase in alignment quality over that of the SmA$_F$ or some electrically modified order parameter effect[56]. When the field is reduced the increased birefringence is retained to lower voltages than the threshold for tilt removal indicating a hysteresis in the tilt removal (Fig. 5c). For analogous measurements on the N$_{TBF}$ phase of **2** the tilt removal is threshold-less, despite the presence of an alignment layer, and has no evidence of hysteresis (Fig. 5d). The threshold-less nature of the change in tilt is due to the avoidance of bound charges that would form in the bulk. Instead, the tilt is maintained homogenous across the active area and bound charges are expelled to the electrodes, similar to director oscillations in the N$_F$ phase[57]. The obtained birefringence is slightly higher than the N$_F$ phase however, not as significant as for the SmC$_P^H$ phase.

The helix in the SmC$_P^H$ and N$_{TBF}$ phases allows for partial polarisation compensation and so removal of the tilt results in an increase in polarisation. A ferroelectric phase has two stable states at $E = 0$, while an anti-ferroelectric phase would only have a single stable state[58]. A ferrielectric phase has two stable states at $E = 0$, but two separate sub-lattices of non-equal polarisation[59]. These sub-lattices can be reversed going through a field-induced phase transition to the ferroelectric state at high electric field. As a result a ferrielectric phase has three hysteresis loops in its P-E curve. The SmC$_P^H$ phase has three separate hysteresis components (polarisation reversal and then tilt hysteresis in both the positive and negative polarisation states) and so has a pseudo-ferrielectric switching response. However, it is not a true ferrielectric phase as its ground state is ferroelectric like with all of its constituent dipoles orientated along the same direction. The N$_{TBF}$ phase has the same ground state as the SmC$_P^H$ but does not exhibit

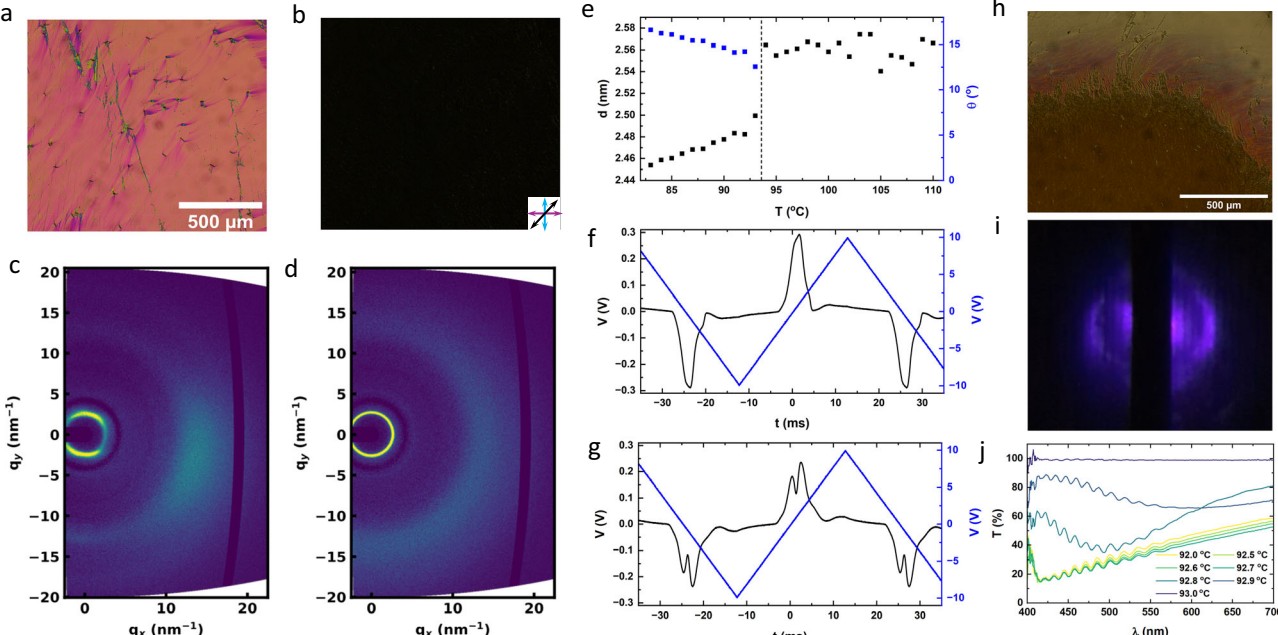

**Fig. 6 | Physical properties of compound 3.** POM images in the **a** $N_F$ and **b** $SmC_P^H$ phases of **3** in a 5 μm planar aligned parallel rubbed cell. Arrows indicate direction of polarisers and rubbing direction. 2D X-ray scattering images of **3** in the **c** $N_F$ and **d** $SmC_P^H$ phases. **e** Temperature dependence of the d-spacing and molecular tilt of **3**. Current responses for **3** in the **f** $N_F$ and **g** $SmC_P^H$ phases. **h** Microscopy image of **3** with the polarisor inserted parallel to the rubbing direction (no analyser) of the boundary between the $N_F$ phase (top) and $SmC_P^H$ phase (bottom) showing selective reflection in the boundary region. **i** diffraction pattern from the boundary region demonstrating a pitch of ~800 μm. **j** Transmission spectrum from the boundary region of image (**a**).

hysteresis in its tilt removal and so only exhibits ferroelectric switching with an additional pseudo-paraelectric component at higher electric fields. We note that this pseudo-ferrielectricity is reminiscent of the concept of reducible and irreducible ferrielectricity in solid state materials[60].

Beyond the $N_{TBF}$ phase, current responses consisting of a small peak pre-polarity reversal and a subsequent larger peak have been observed for the non-helical $SmC_P$ phase[37,39,61], though this is apparently not always observed[38]. This peak has been associated with tilt reformation and indicates the possibility of a pseudo-ferrielectric responses for the the non-helical $SmC_P$ phase.

**Pre-transitional helix formation**

We have demonstrated how variation of the phase ordering from $SmC_P^H - N_{TBF} - N_F$ to $SmC_P^H$-$SmA_F$ varies the properties of the $SmC_P^H$ phase. Both of the aforementioned transitions to heliconic order are 2nd order indicated by their lack of enthalpy from DSC and their continuous, rather than discontinuous, tilt dependence at the transition to heliconic order (Fig. S11). Variation in fluorination is a now well established technique to obtained various polar phases[30,42,62] and so changing the chemical structure to that of compound **3** (Table 1) results in a phase order of $N_F - SmC_P^H$. POM observations reveal the $SmC_P^H - N_F$ transition on cooling occurs as a wave edged with dendritic tendrils (Fig. 6a, b). The transition front moves across the cell, forming a similar scattering texture as observed for **2**, nucleating from specific points across the cell. This is contrary to the smooth uniform change in texture as observed for the $N_{TBF}$–$N_F$ transition of **2**. On heating the scattering texture melts with a sharp boundary between $SmC_P^H$ and $N_F$ domains.

X-ray scattering measurements show a large jump in tilt angle formation up to ~12° molecular tilt immediately at the transition, indicating a strong first-order character (Fig. 6c–e). We also observe a comparatively large enthalpy compared to the lack of enthalpy for the 2nd order transitions $SmC_P^H$-$SmA_F$ and $N_{TBF}$–$N_F$ as well as the 1st order $SmC_P^H - N_{TBF}$ transition. All of this points to the $SmC_P^H - N_F$ transition

of **3** as being the first 1st order transition to spontaneous heliconic order. Current response measurements show a single polarisation reversal peak for the $N_F$ phase (Fig. 6f). Upon transition into the $SmC_P^H$ phase the peak splits into two with a smaller tilt peak at pre-polarity voltages and larger polarisation reversal peak (Fig. 6g), which moves to longer time scales with reducing temperature. The larger initial size of the tilt peak is owed to the first-order nature of the transition leading to an discontinuous large change in the molecular tilt and so a larger change in polarisation upon tilt reformation.

If the boundary between the $N_F$ and $SmC_P^H$ domains is observed on cooling, a narrow region of ~300 μm width exists where selective reflection can be seen (Fig. 6h). Across this boundary layer the colour varies from red to purple, indicating an increasing in bandgap wavelength from the $SmC_P^H$ domain to the $N_F$ domain. Measuring the wavelength of the selective reflection directly shows it does indeed shift rapidly though the visible window in 0.2 °C indicating rapid change of the helical pitch in this region (Fig. 6j). Direct remeasurements of the pitch via laser diffraction confirm this indicating a change in pitch from 920 μm to less than 600 μm within that same 0.2 °C window (Fig. 6i). We considered whether this is a narrow temperature window of $N_{TBF}$ phase existence, however on heating the region of selective reflection and light diffraction due to the helical pitch is not observed (Supplementary Videos 1 and 2). We interpret this as either the $SmC_P^H$ phase front growing across the cell and forming initially in a thin area nearer the base due to any vertical temperature gradients, which then produces the selective reflection and pitch. Alternatively, this is a pre-transitional effect where the helix in the $SmC_P^H$ acts in induce some helical ordering in the neighbouring $N_F$ regions. As this is not the lowest free energy state for the $N_F$ phase this helix unwinds as distance from the phase front is increased until it smoothly merges with the non-helical $N_F$ regions.

Further changes in the fluorination pattern results in compound **4**, however, this results in a destabilisation of any smectic or helical phases and instead only an enantiotropic $N_F$ phases is observed. We also note the strong destabilisation of the apolar-polar transition by ~

20 °C. This is explained as a decreased spatial uniformity of the alternating charge regions across the molecule preventing reducing the strength of the lateral interactions that promote polar phases[25,30].

## Conclusions

The discovery of spontaneous chiral symmetry breaking in the ferroelectric nematic realm[36,41] is extremely exciting due to the potential for replicating the behaviours seen in hard matter systems such as ferromagnets in fluid soft matter systems. Here we have demonstrated that the $SmC_F$ phase found in **2**[41] is helical and is in fact a $SmC_P^H$ phase, as observed for **1**[36]. We show that there is at least three phase pathways to the $SmC_P^H$ (we expect that there could well be more pathways then the three presented here). Namely $SmC_P^H$-$SmA_F$, $SmC_P^H - N_{TBF}$ and $SmC_P^H - N_F$ and that each leads to an array of different polymorphs and interesting properties.

We account for the double peak structure of the current response for the $SmC_P^H$ phase, demonstrating that the smaller pre-polarity peak is associated with reformation of the tilt and show how the behaviour of this tilt peak is influenced by the phase preceding it. The tilt peak of the $SmC_P^H$ phase is indicative of field-induced phase transitions to the $SmA_F$ state. This phase transition shows clear hysteresis, indicating that it is pseudo-ferrielectric due to the increase in polarisation that occurs by removing molecular tilt from a partially compensated helical phase.

Looking forward, spontaneous symmetry breaking due to polar order may yield interesting new phase types and states when superimposed on more complex liquid crystal organisations, for example, the twisted polar smectic structures described recently by Nishikawa et al.[46].

## Methods

### Chemical synthesis

All chemicals and solvents were purchased from commercial suppliers and used as received. Reactions were performed in standard laboratory glassware at ambient temperature and atmosphere and were monitored by TLC with an appropriate eluent and visualised with 254 nm light. Chromatographic purification was performed using a Combiflash NextGen 300+ System (Teledyne Isco) with a silica gel stationary phase and a hexane/ethyl acetate gradient as the mobile phase, with detection made in the 200–800 nm range. Full details are given in the Supplementary Information.

### Thermal analysis

DSC measurements were performed using a TA Instruments Q2000 DSC instrument (TA Instruments, Wilmslow UK), equipped with a RCS90 Refrigerated cooling system (TA Instruments, Wilmslow, UK). The instrument was calibrated against an Indium standard, and data were processed using TA Instruments Universal Analysis Software. Samples were analysed under a nitrogen atmosphere, in hermetically sealed aluminium TZero crucibles (TA Instruments, Wilmslow, UK).

### Optical microscopy

Polarised light optical microscopy (POM) was performed using a Leica DM2700P polarised light microscope (Leica Microsystems (UK) Ltd., Milton Keynes, UK), equipped with 10× and 50× magnification, and a rotatable stage. A Linkam TMS 92 heating stage (Linkam Scientific Instruments Ltd., Redhill, UK) was used for temperature control, and samples were studied sandwiched between two untreated glass coverslips. Images were recorded using a Nikon D3500 Digital Camera (Nikon UK Ltd., Surbiton, UK), using DigiCamControl software.

### X-ray scattering

X-ray scattering measurements, both small angle (SAXS) and wide angle (WAXS), were recorded using an Anton Paar SAXSpoint 5.0 beamline machine. This was equipped with a Primux 100 Cu X-ray source with a 2D EIGER2 R detector. The X-rays had a wavelength of 0.154 nm. Samples were filled into either thin-walled quartz capillaries or held between Kapton tape.

### Current response

Spontaneous polarisation measurements were undertaken using the current reversal technique, using a Agilent 33220A signal generator and a RIGOL DHO4204 high-resolution oscilloscope, with temperature control via an Instec HCS402 hot stage.

### Birefringence measurements

Birefringence was measured using a Berek compensator mounted in a Leica DM 2700 P polarised optical microscope. The birefringence was measured in the centre of the active ITO area for the IPS cells used here. Error bars were assessed by repeating the measurement several times and using the average to obtained a statistical error.

### Transmission spectra

A microscope equipped coupled to an Avantes AvaSpec-2048 XL spectrometer, which used to record transmission spectra of samples. The rubbing direction was set parallel to the light polarization direction with no analyser inserted. A spectra from the phase above the spontaneously chiral phases was used as the reference state of each sample.

### Pitch measurements

The pitch was measured by illumination of the sample from the bottom via a 405 nm laser. The angle of diffraction was measured on a flat screen set 1.8 cm away from the sample.

## Data availability

The data associated with this paper are openly available from the University of Leeds Data Repository at: https://doi.org/10.5518/1643. Source data are provided with this paper.

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

## Acknowledgements

R.J.M. thanks UKRI for funding via a Future Leaders Fellowship, grant number MR/W006391/1, and the University of Leeds for funding via a University Academic Fellowship. The SAXS/WAXS system used in this work was funded by EPSRC via grant number EP/X0348011. R.J.M. gratefully acknowledge support from Merck KGaA. Computational work was performed on ARC4 and AIRE, part of the high-performance computing facilities at the University of Leeds.

## Author contributions

J.H. performed the experiments, conducted the analysis and wrote the first draft. C.J.G. synthesised the material and collected chemical characterisation data. R.J.M. secured funding. All authors contributed to reviewing and editing the final manuscript.

## Competing interests

The authors declare no competing interests.
