## [Transparent Peer Review file · Nature Communications]

Ferri- and Ferro-Electric Switching in Spontaneously Chiral Polar Liquid Crystals

Corresponding Author: Dr Jordan Hobbs

Version 0:

Reviewer comments:

Reviewer #1

(Remarks to the Author)

This paper seems to be a follow up paper of references 29 (Nature Communications 15, 5845 (2024)) and 34 (Science 384, 1096 (2024)) with aim to find out whether the SmCF phase reported in Ref. 34 and the SmCPH reported in 29 are the same or not.

The authors synthesize and mix the substances where these phases were observed, and find complete miscibility, which is a strong indication for the answer that SmCF and SmCPH.

The second major concern of the paper is the nature of the ferroelectric like double polarization peaks during electric switching. The smaller peak appears in decreasing fields, the larger one in increasing fields. They claim the smaller one is due to director tilt, while the other one is the "polarization reversal current" during a field-induced SmCF-SmAF transition. As a tilt angle alters the layer spacing, such a transition would require formation of new layers. As smectic layers cannot be generated readily, it would lead to layer tilt (chevron or striped bookshelf structure formation) that generate defects. I doubt it could be done reversibly in 20 ms time scale.

I am wondering why the authors do not assume simple helix unwinding during the "polarization reversal" and helix reformation instead of "tilt" process during the peak in decreasing fields.

In general, it is very difficult to read the paper and follow the arguments just verbally without structural cartoons that would clearly indicate what structural changes happen during the peaks of the electro-optical transmissions an electric current shown in Fig.2 and in the POM texture formations shown in Figure 3-6.

In my opinion, although the paper is interesting and a more readable version with well-illustrated structural cartoons will be very important for the LC community dealing with the ferroelectric nematic realm. However, it lacks the broader interest factor, and it might be more appropriate for a more specialized journal such as Liquid Crystals.

Reviewer #2

(Remarks to the Author)

The manuscript presents an analysis of the newly synthesized materials with ferroelectric ordering; it belongs to the explosively growing field of ferroelectric nematic "realm". The authors demonstrates that the smectic C polar (ferroelectric) phase (SmCF) that forms below the ferroelectric twist-bend nematic phase is a phase with a helicoidal structure, abbreviated as SmCPH, as opposed to non-helical SmCP. The work employs a battery of modern optical, electro-optical and electric probing techniques to characterize the materials and the conclusions are justified. The results should be of interest to a broad community of researchers in the field of liquid crystals but also of solid ferroelectrics, as the search of chiral states in these is currently underway. The manuscript can be accepted after the authors consider the following issues:

1. The text should be carefully checked since in some places the sentences are incomplete (for example, in Section VC, "The transition to the from..." and "a field induced phase transition the untitled state". The statement "The NF phase can spontaneously break chiral symmetry to form pi-twisted domains [25]" is not accurate since in [25], the pi-twists naturally occur in planar cells assembled in an antiparallel fashion; there is no twist in parallel assembly; such a twist in the antiparallel cells is imposed by the surface anchoring rather than forms "spontaneously".
2. What is the nature of NTBF stripes parallel to the rubbing direction?
3. How was the birefringence measured? How were the error bars for birefringence determined?

Reviewer #3

(Remarks to the Author)

Comments to the authors:

The authors report on the SmCPF of compound 2, which appears to adopt a helical structure similar to the heliconical phase previously reported for Compound 1. The authors analysed four compounds, two of which are reported before. The paper could only be accepted for publication in Nature Communication after major revision. See my comments below.

1. Could the authors provide soft-resonant X-ray investigations to prove the helix formation in the discussed phase? Also, SHG and dielectric studies would be helpful to prove the nature of the polar phase, if it is really ferroelectric or antiferroelectric.
2. What is the Nx phase given in Fig. 1j? What are the characteristic features of this phase?
3. What is the main idea of reporting compound 4? I do not see any discussion about this material in the manuscript or the in the SI.
4. In Fig. 7, please indicate which compound is investigated.
5. I am missing the DSC thermograms of the new materials. Please provide them in the SI.
6. There are many typos which should be checked carefully and corrected. Also please check the correct figure assignments in the main text, when it is referring to those in the SI.
7. Concerning the SI file: Where is the synthesis scheme 1? What is the percent yield for each phenolic compound following this scheme? There is also extra text "4 Hydroxy" given before the NMR data of the first phenol after Scheme 1.
8. The authors might like to cite the recent work about heliconical smectic phases reported very recently: DOI: 10.1039/D5TC01109D

Version 1:

Reviewer comments:

Reviewer #1

(Remarks to the Author)

The authors have adequately addressed my questions and comments and revised the paper accordingly. I suggest acceptance as is now.

Reviewer #2

(Remarks to the Author)

The revised manuscript answers my comments except one. The authors state that "the NF phase can form pi-twisted domains [31] which can occur spontaneously even with degenerate anchoring orientations [27, 32] to reduce the electrostatic cost of polarization [27, 33]." One point here is not accurate: Yadav et al. in Ref. 32 report on spontaneous twists in the high-temperature conventional paraelectric nematic (N) phase rather than in the ferroelectric nematic (NF) phase. The authors of Ref.32 stress that the twisted domains in the NF phase form only when the two uniformly rubbed plates are assembled in an antiparallel fashion, see Fig.9 in Ref. 32 and the associated text. The conclusions of Refs. 32 and 27 are different: Ref. 27 claims spontaneous twists for the NF phase and not for the N phase, while Ref. 32 claims spontaneous twists for the N phase and not the NF phase. After the statement is edited, the manuscript can be accepted for publication.
Signed: Oleg D. Lavrentovich

Reviewer #3

(Remarks to the Author)

The authors have addressed all minor points satisfactorily. However, they did not provide the requested data from soft-resonant X-ray and dielectric measurements, which were the major points. These additional experimental results are essential to support the claims of this study with robust evidence. I believe that publishing in Nature Communication needs a clear and solid prove of the new data, which is not the case for the current work without the mention data. Therefore, I cannot recommend the publication of the manuscript in Nature Communication.

Dr. Jordan Hobbs
Sir William Henry Bragg Building
University of Leeds
Leeds, UK, LS2 9JT

RE: "Ferri- and Ferro-Electric Switching in Spontaneously Chiral Polar Liquid Crystals"
NCOMMS-25-15422

We would like to thank the editorial staff and the referees for their time and reviews. In response we provide a revised version which we believe addresses all the comments and suggestions. In both the manuscript and ESI changes are marked in red text.

REVIEWER COMMENTS

Reviewer #1 (Remarks to the Author):

This paper seems to be a follow up paper of references 29 (Nature Communications 15, 5845 (2024)) and 34 (Science 384, 1096 (2024)) with aim to find out whether the SmCF phase reported in Ref. 34 and the SmCPH reported in 29 are the same or not. The authors synthesize and mix the substances where these phases were observed, and find complete miscibility, which is a strong indication for the answer that SmCF and SmCPH.

This is certainly one of the aims of the paper. Having clear, accurate and unified terminologies for these rapidly emerging phases is extremely important for the wider audience to understand this new body of work.

The second major concern of the paper is the nature of the ferrielectric like double polarization peaks during electric switching. The smaller peak appears in decreasing fields, the larger one in increasing fields. They claim the smaller one is due to director tilt, while the other one is the "polarization reversal current" during a field-induced SmCF-SmAF transition. As a tilt angle alters the layer spacing, such a transition would require formation of new layers. As smectic layers cannot be generated readily, it would lead to layer tilt (chevron or striped bookshelf structure formation) that generate defects. I doubt it could be done reversibly in 20 ms time scale.

We agree that we expected chevron defects however we simply do not observe them and the data shows that we are indeed altering the layer spacing. We see this as part of the impact of our work. The P-E coupling appears to be strong enough to reorientate the dipoles to be fully perpendicular to the electrodes. Upon removal of the field the tilt would likely reform much as it does upon its natural formation under temperature reduction which clearly does not result in chevron formation due to the absence of defects associated with those structures. We also note that since submitting this manuscript, a publication by the Warsaw group has come out demonstrating the same effect here using light diffraction measurements (Gorecka et.al. 2025, Mater. Horiz.). They also see no evidence for

chevron defects in their samples. We do stress that neither papers elucidate the dynamics within the switching process which could be extremely interesting as well.

I am wondering why the authors do not assume simple helix unwinding during the “polarization reversal” and helix reformation instead of “tilt” process during the peak in decreasing fields.

We do already consider this within the paper itself at the beginning of section VA:

“The peak cannot be due to variation in the helical pitch length as this would not modulate the polarisation beyond a small variation due to changes in the region of partial pitch that may form due to the degenerate anchoring condition of the bare ITO electrode. Variation in the tilt angle, however, would modulate the effective polarisation observed upon switching.”

Since the polarisation axis lies parallel to the director, the heliconic structure acts to partially compensate for the polarisation. The more tilted the molecules are the larger this compensation effect. However, modulation of the helical pitch itself does not affect the projection of the molecular dipole axis along the director and so will not lead to variations in the polarisation. Of course, by removing the tilt completely as we show occurs here, the helix must fully unwind but the unwinding itself is not the source of the small peak in the current response due to the above stated reasons.

In general, it is very difficult to read the paper and follow the arguments just verbally without structural cartoons that would clearly indicate what structural changes happen during the peaks of the electro-optical transmissions an electric current shown in Fig.2 and in the POM texture formations shown in Figure 3-6.

We appreciate the reviewer’s suggestion of structural cartoons and so have included some in appropriate places to add in interpretation of our arguments. We believe these should make our written arguments clear.

In my opinion, although the paper is interesting and a more readable version with well-illustrated structural cartoons will be very important for the LC community dealing with the ferroelectric nematic realm. However, it lacks the broader interest factor, and it might be more appropriate for a more specialized journal such as Liquid Crystals.

We thank the reviewer for their positive views of our work, though we of course disagree about the relevancy to the wider scientific community. Spontaneously chiral ferroelectric structures are rare in solid state ferroelectrics and these materials provides an opportunity to understand how these phenomena can occur in the absence of a restrictive crystal lattice. Understanding the switching processes is paramount to understanding the structure appearance of these new phases. We would like to note that there are some parallels between discussions around reducible and irreducible ferrielectricity that is discussed in the solid state and so we expect there to be genuine interest from the wider community. We also highlight the positive comments by the other reviewers who share our view on the appeal of this work to the broader scientific community.

Reviewer #2 (Remarks to the Author):

The manuscript presents an analysis of the newly synthesized materials with ferroelectric ordering; it belongs to the explosively growing field of ferroelectric nematic “realm”. The authors demonstrates that the smectic C polar (ferroelectric) phase (SmCF) that forms below the ferroelectric twist-bend nematic phase is a phase with a helicoidal structure, abbreviated as SmCPH, as opposed to non-helical SmCP. The work employs a battery of modern optical,

electro-optical and electric probing techniques to characterize the materials and the conclusions are justified. The results should be of interest to a broad community of researchers in the field of liquid crystals but also of solid ferroelectrics, as the search of chiral states in these is currently underway.

We appreciate the reviewers positive view of our work.

The manuscript can be accepted after the authors consider the following issues:

1. The text should be carefully checked since in some places the sentences are incomplete (for example, in Section VC, “The transition to the from...” and “a field induced phase transition the untitled state”.

We thank the reviewer for their thorough reading of our manuscript, and we have fixed these (and other) typos in the new version.

The statement “The NF phase can spontaneously break chiral symmetry to form pi-twisted domains [25]” is not accurate since in [25], the pi-twists naturally occur in planar cells assembled in an antiparallel fashion; there is no twist in parallel assembly; such a twist in the antiparallel cells is imposed by the surface anchoring rather than forms “spontaneously”.

We agree that this statement could be better written. It was actually work by Kumari et. al. that showed that this twist will occur even in degenerate anchoring conditions and as such could be considered as spontaneous twisting. We do cite this in the next sentence, but it is not clear that this is the work establishing the spontaneity of the twist. We note that it was also shown by the Dublin group in 2023 and since these two works other groups have also demonstrated the spontaneity on the twist.

“It has been shown that the N_F phase can form π -twisted domains [31] which can occur spontaneously even with degenerate anchoring orientations [27, 33] to reduce the electrostatic cost of polarisation [27, 33].”

2. What is the nature of NTBF stripes parallel to the rubbing direction?

We are not certain at this point. As we discuss in the manuscript, we know these are not pitch defects or markers or alternating regions of polarity. We do have our suspicions about them being related to Helfrich-Hurault instabilities as an analogy to the stripe defects of the twist bend nematic phase in flexible dimers, but we do not have any evidence to suggest this currently and would prefer to avoid speculation with no evidence. We expect that a thorough explanation of these stripes will be a significant publication in future by whoever can provide a sufficient model.

3. How was the birefringence measured? How were the error bars for birefringence determined?

The birefringence was measured using a Berek Compensator. While we included this detail in the SI we did not really make that clear in the main manuscript. We have now changed this and added detail about the error bars which are simply the distribution of multiple measurements.

Reviewer #3 (Remarks to the Author):

Comments to the authors:

The authors report on the SmCPF of compound 2, which appears to adopt a helical structure similar to the heliconical phase previously reported for Compound 1. The authors analysed four compounds, two of which are reported before. The paper could only be accepted for publication in Nature Communication after major revision. See my comments below.

1. Could the authors provide soft-resonant X-ray investigations to prove the helix formation in the discussed phase? Also, SHG and dielectric studies would be helpful to prove the nature of the polar phase, if it is really ferroelectric or antiferroelectric.

We would love to provide RSoX measurements however these measurements require application and acceptance for beam time at specialist facility. We consider these types of measurements to be extremely interesting but are beyond the scope of this paper. SHG measurements of compounds **1** and **2** has already been published in Nature Communications 15, 5845 (2024) and Science 384, 1096 (2024) respectively. Interpretation of dielectric spectroscopy of these highly polar and fluid materials is currently extremely contentious and so we would prefer to avoid publishing this data without a full analysis and discussion which we expect to be a full publication in its own right. The polar nature of these phases is indeed an interesting question. We consider the SmCHP phase to have a ferroelectric ground state as all the dipoles are orientated in a single direction. The switching profile is ferrielectric like (which itself can be considered a special type of anti-ferroelectricity with non-equal lattices) but crucially the ground state is only a single lattice, so we consider it to be ferroelectric with pseudo-ferrielectric like switching properties.

2. What is the N_x phase given in Fig. 1j? What are the characteristic features of this phase?

The N_x phase is an antiferroelectric nematic phase. Due to some disagreements over the specifics of the structure of this phase it is sometimes referred to as the N_S , SmZ_A , N_{AF} or M_2 phase. We prefer the usage of N_S but used N_x in the figure by mistake. This has been corrected in the updated version of this manuscript.

3. What is the main idea of reporting compound 4? I do not see any discussion about this material in the manuscript or the in the SI.

The main idea is to show the sensitivity of these new helical phases to chemical structure. So far, these seem to be even more sensitive to molecular structure than the N_F phase is. We have included a brief paragraph in describe this compound at the end of the manuscript.

“Further changes in the fluorination pattern results in compound 4, however this results in a destabilisation of any smectic or helical phases and instead only an enantiotropic N_F phases is observed. We also note the strong destabilisation of the apolar-polar transition by ~ 20 °C. This is explained as a decreased spatial uniformity of the alternating charge regions across the molecule preventing reducing the strength of the lateral interactions that promote polar phases [25, 30].”

4. In Fig. 7, please indicate which compound is investigated.

This was about compound 3 and has been corrected in the caption.

5. I am missing the DSC thermograms of the new materials. Please provide them in the SI.

We have included these in the SI as figure S14

6. There are many typos which should be checked carefully and corrected. Also please check the correct figure assignments in the main text, when it is referring to those in the SI.

Thank you for the thorough review of our manuscript. We have given the updated version a thorough check and believe we have now caught these typos and misassigned figures.

7. Concerning the SI file: Where is the synthesis scheme 1? What is the percent yield for each phenolic compound following this scheme? There is also extra text "4 Hydroxy" given before the NMR data of the first phenol after Scheme 1.

We have gone through and corrected/included the missing chemical characterisation data in the SI

8. The authors might like to cite the recent work about heliconical smectic phases reported very recently: DOI: 10.1039/D5TC01109D

Thank you for this suggestion. We have included this in the introduction to help with our brief illustration of the breadth of chiral symmetry breaking in LC phases.

Once again, we would like to thank the editor and reviewers for their review of our manuscript, and we look forward to hearing the outcome of this revised submission.

Dr. Jordan Hobbs

Research Fellow

@: j.l.hobbs@leeds.ac.uk

Dr. Jordan Hobbs
Sir William Henry Bragg Building
University of Leeds
Leeds, UK, LS2 9JT

RE: "Ferri- and Ferro-Electric Switching in Spontaneously Chiral Polar Liquid Crystals"
NCOMMS-25-15422

We would like to thank the editorial staff and the referees for their time and reviews. In response we provide a final version which we believe addresses all the comments and suggestions.

REVIEWER COMMENTS

Reviewer #1 (Remarks to the Author):

The authors have adequately addressed my questions and comments and revised the paper accordingly.

I suggest acceptance as is now.

We thank the reviewer for their time and comments throughout this review process.

Reviewer #2 (Remarks to the Author):

The revised manuscript answers my comments except one. The authors state that "the NF phase can form pi-twisted domains [31] which can occur spontaneously even with degenerate anchoring orientations [27, 32] to reduce the electrostatic cost of polarization [27, 33]." One point here is not accurate: Yadav et al. in Ref. 32 report on spontaneous twists in the high-temperature conventional paraelectric nematic (N) phase rather than in the ferroelectric nematic (NF) phase. The authors of Ref.32 stress that the twisted domains in the NF phase form only when the two uniformly rubbed plates are assembled in an antiparallel fashion, see Fig.9 in Ref. 32 and the associated text. The conclusions of Refs. 32 and 27 are different: Ref. 27 claims spontaneous twists for the NF phase and not for the N phase, while Ref. 32 claims spontaneous twists for the N phase and not the NF phase. After the statement is edited, the manuscript can be accepted for publication.

Signed: Oleg D. Lavrentovich

Upon reviewing reference 32 again we realise that Professor Lavrentovich is indeed correct about it being restricted to only the N phase. We have removed the reference from the final manuscript.

We thank the reviewer for their comments and time contributed to this review process.

Reviewer #3 (Remarks to the Author):

The authors have addressed all minor points satisfactorily. However, they did not provide the requested data from soft-resonant X-ray and dielectric measurements, which were the major points. These additional experimental results are essential to support the claims of this study with robust evidence. I believe that publishing in Nature Communication needs a clear and solid prove of the new data, which is not the case for the current work without the mention data. Therefore, I cannot recommend the publication of the manuscript in Nature Communication

While we appreciate the reviewers request for additional data, these techniques are neither unambiguous nor uncontentious and fall outside the scope of this present work.

We thank the reviewer for their suggestions and comments throughout this review process.

Dr. Jordan Hobbs

Research Fellow

@: j.l.hobbs@leeds.ac.uk